# Transcriptomic Analysis of Colorectal Cancer Cells Treated with Oil Production Waste Products (OPWPs) Reveals Enrichment of Pathways of Mitochondrial Functionality

**DOI:** 10.3390/cells11243992

**Published:** 2022-12-10

**Authors:** Manuela Leo, Livio Muccillo, Erica Pranzini, Giovannina Barisciano, Matteo Parri, Giulia Lopatriello, Marco Carlomagno, Alice Santi, Maria Letizia Taddei, Lina Sabatino

**Affiliations:** 1Department of Sciences and Technologies, University of Sannio, Via Francesco de Sanctis, 82100 Benevento, Italy; 2Department of Experimental and Clinical Biomedical Sciences, University of Florence, Viale Morgagni 50, 50134 Firenze, Italy; 3Department of Biotechnology, University of Verona, Strada le Grazie 15, Cà Vignal 1, 37134 Verona, Italy; 4Department of Experimental and Clinical Medicine, University of Florence, Viale Morgagni 50, 50134 Firenze, Italy

**Keywords:** oil production waste products (OPWPs), hydroxytyrosol, polyphenols, mitochondrial biogenesis, mitochondrial dynamics, PPARγ, PGC-1α, colorectal cancer

## Abstract

Oil production waste products (OPWPs) derive from olive mill and represent a crucial environmental problem due to their high polyphenolic content able to pollute the ground. One option to reduce the OPWPs’ environmental impact is to exploit polyphenols’ biological properties. We sought to analyze the transcriptomic variations of colorectal cancer cells exposed to the OPWPs extracts and hydroxytyrosol, the major component, to recognize unknown and ill-defined characteristics. Among the top affected pathways identified by GSEA, we focused on oxidative phosphorylation in an in vitro system. Colorectal cancer HCT116 and LoVo cells treated with hydroxytyrosol or OPWPs extracts showed enhancement of the respiratory chain complexes’ protein levels, ATP production and membrane potential, suggesting stimulation of mitochondrial functions. The major proteins involved in mitochondrial biogenesis and fusion events of mitochondrial dynamics were positively affected, as by Western blot, fostering increase of the mitochondrial mass organized in a network of elongated organelles. Mechanistically, we proved that PPARγ mediates the effects as they are mimicked by a specific ligand and impaired by a specific inhibitor. OPWP extracts and hydroxytyrosol, thus, promote mitochondrial functionality via a feed-forward regulatory loop involving the PPARγ/PGC-1α axis. These results support their use in functional foods and as adjuvants in cancer therapy.

## 1. Introduction

Oil production waste products (OPWPs) is the term used to indicate oil-free olive pulp (solid) and olive mill wastewaters (liquid) derived from the olive washing process and olive oil purification procedures. These products are unfortunately associated with severe environmental concerns [1,2], because of the high content of polyphenols that infiltrate and pollute the ground due to their low biodegradability [3]. The control of this environmental problem is an emerging issue along with the development of sustainable processes aimed at exploiting such wastes [4,5,6]. The most abundant phenolic compound in OPWPs is hydroxytyrosol (HTyr), which is also present in olive oil and olive leaf extracts [7]. It derives from the precursor oleuropein that, through the olive milling process, is modified by endogenous β-glycosidases to the final polar compound. Despite the detrimental effects on the environment, the entire class of polyphenols display antioxidant, anti-inflammatory, antiproliferative, and hypoglycemic properties following purification or enrichment procedures [7,8,9,10,11].

Exposure to polyphenol extracts induces, in most cases, changes in cellular metabolism [12,13]. The term metabolic regulation designates the processes by which mammals regulate catabolic/degradative pathways that involve the oxidization of available substrates to produce energy necessary to support anabolic/biosynthetic activities and functions [14,15]. Mitochondria are the main site for energy production, as ATP is strictly associated with mitochondrial homeostasis. Hormonal and metabolic signals tightly regulate the response to energy-consuming conditions by modifying the mitochondrial metabolism either by finely tuning the amount/activity of mitochondrial enzymes and regulatory proteins and/or modifying the number and the shape of mitochondria through events defined as biogenesis and dynamics [16].

Mitochondria biogenesis is the means whereby mitochondria self-replicate [17]. In contrast, mitochondria dynamics involves modifications in shape and interactions, proceeding from single punctuated to elongated organelles organized in a network [18,19]. The master regulator of mitochondrial structure and function is Peroxisome Proliferator Activated Receptor Gamma Coactivator 1α (PGC-1α) [20,21]. It cooperates with many other factors to regulate genes involved in mitochondrial metabolism, specifically with Peroxisome Proliferator Activated Receptor γ (PPARγ), a ligand-dependent transcription factor with major roles in lipid and energetic metabolism [22,23].

In this study, we sought to examine the transcriptional changes occurring in colorectal cancer (CRC) cells following the treatment with OPWPs extracts or the major constituent HTyr. We focused on Oxidative Phosphorylation (OXPHOS) since it was among the top four most differentially regulated pathways. We show that OPWPs extracts and HTyr greatly affect mitochondrial respiration and mass by promoting biogenesis and dynamics. Mechanistically, these effects are mediated by the PPARγ and PGC-1α axis through a feed-forward regulatory loop. Thus, OPWPs extracts and HTyr promote mitochondrial functionality that adds to their antiproliferative and proapoptotic activities [7,8,9,10,11]. All these characteristics support their use as adjuvant anticancer drugs, nutraceuticals, additives in cosmetics and foodstuffs.

## 2. Materials and Methods

### 2.1. OPWPs Extracts 

Hydroxytyrosol (HTyr) was synthesized according to the already described procedure [24]. The three HTyr-enriched OPWPs extracts were produced at different sites across Italy, namely Apulia, Tuscany and Sicily, henceforth named AE (Apulia Extract), TE (Tuscany Extract) and SE (Sicily Extract), respectively. They were characterized as previously reported [25].

### 2.2. Cell Culture and Treatments

The human CRC-derived cell lines, HCT116 and LoVo, were acquired from the American Type Culture Collection (ATCC, Manassas, VA, USA) and cultured at 37 °C under 5% CO_2_ in RPMI 1640 (Life Technologies, Carlsbad, CA, USA) supplemented with 10% *v*/*v* fetal bovine serum (Life Technologies, Carlsbad, CA, USA), 2 mM L-glutamine, 100 U mL^−1^ penicillin/streptomycin. Oil production waste products (OPWPs) freeze-dried extracts were resuspended in sterile water at the concentration of 100 mg mL^−1^ (corresponding to their solubility). In all experiments, HCT116 cells were treated with 0.0154 mg mL^−1^ HTyr or 0.0154 mg mL^−1^ OPWPs extracts for 72 h; LoVo cells were treated with 0.0231 mg mL^−1^ HTyr or 0.0231 mg mL^−1^ OPWPs extracts for 72 h. These values were obtained after assessment of the IC_50_ for HTyr in each cell line [25]. In some experiments, HCT116 and LoVo cells were treated with rosiglitazone (1 µM), a PPARγ-selected agonist or with GW9662 (10 µM), a PPARγ irreversible antagonist, for 72 h (both from Cayman, Ann Arbor, MI, USA) dissolved in dimethyl sulfoxide (DMSO, Merck-Sigma, St. Louis, MO, USA); DMSO alone was used as a vehicle control.

### 2.3. RNA Extraction and Sequencing 

Total RNA was extracted from cells in culture with Trizol (Invitrogen, Carlsbad, CA, USA), following the manufacturer’s instructions. RNA quality and integrity controls were assessed using the RNA 6000 Nano Kit on a Bioanalyzer (Agilent Technologies, Santa Clara, CA, USA). All samples showed an RNA integrity number (RIN) >8. RNA samples were quantified using the Qubit RNA BR Assay Kit (Thermo Fisher Scientific, Whaltam, MA, USA). RNA-seq libraries were generated using the TruSeq stranded mRNA kit (Illumina, Agilent Technologies, Santa Clara, CA, USA) from 1000 ng of RNA samples, after poly(A) capture and according to manufacturer’s instructions. Quality and size of RNAseq libraries were assessed by capillary electrophoretic analysis with the Agilent 4150 Tape station (Agilent Technologies, Santa Clara, CA, USA). Libraries were quantified by real-time PCR against a standard curve with the KAPA Library Quantification Kit (KapaBiosystems, Wilmington, MA, USA). Libraries were pooled at equimolar concentration and sequenced with Illumina technology analyzing on average 17 million fragments in 150PE mode for each sample.

### 2.4. Bioinformatic Analysis 

FastQC software (http://www.bioinformatics.babraham.ac.uk/projects/fastqc/, accessed on 5 September 2021) was used to assess the quality of reads. Raw reads with more than 10% of undetermined bases or more than 50 bases with a quality score <7 were discarded. Subsequently, reads were clipped from adapter sequences using Scythe software (https://github.com/vsbuffalo/scythe, accessed on 5 September 2021), and low-quality ends (Q score <20 on a 15-nt window) were trimmed with Sickle (https://github.com/vsbuffalo/sickle, accessed on 5 September 2021). Filtered reads were aligned to the Human reference genome GRCh38 using STAR aligner. Gene expression quantification was performed using RSEM and Ensembl v104 annotation. Genes level abundance, estimated counts and gene length obtained with RSEM were summarized into a matrix using the R package tximport and, subsequently, the differential expression analysis was performed with DESeq2. To generate more accurate log2 fold-change estimates for low expressed genes, the shrinkage of the Log_2_ Fold-Change (LFC) was performed applying the apeglm method [26].

Gene Ontology (GO) enrichment analysis of Differential Expressed Genes (DEGs), with *p* adj < 0.05 and LFC > 0.585 or LFC < −0.585, in at least one of the comparisons between treatment and control, were performed using GSEA (https://www.gsea-msigdb.org/gsea/index.jsp, accessed on 12 November 2021). All the enriched Biological Processes and GO categories exceeding the FDR (False Discovery Rate) *p*-value threshold (5%) were selected and plotted. The function simplify was applied to remove redundancy of enriched GO terms.

### 2.5. Western Blot Analysis

Protein extracts were analyzed as previously reported [27]. Briefly, the cells were lysed by an ice-cold lysis buffer containing 25 mM Tris-HCl, pH 7.5, 150 mM NaCl, 2 mM EDTA, 1% Triton X-100, 1% sodium deoxycholate, 0.1% SDS, a cocktail of protease and phosphatase inhibitors (Roche, Basel, Switzerland). Thirty µg of total protein extracts were heated at 95 °C for 5 min and loaded on reducing polyacrylamide gels, transferred onto a PVDF membrane, subsequently blocked with 5% non-fat dry milk. Antibodies to PPARγ (sc-7273) and PGC-1α (sc-517380) were from Santa Cruz Biotechnology (Dallas, TX, USA); TOM 20 (#42406), MFN1 (#14739), MFN2 (#11925), MFF (#84580) from Cell Signaling Technology (Beverly, MA, USA); β-Actin (F-3022) and α-Tubulin (T-5168) from Merck-Sigma, (St. Louis, MI, USA), FOXJ3 (A303-107A-T), TFAM (#A303-226A-T) and FOXJ3 (#A303-107A-T) from BETHYL Laboratories (Montgomery, TX, USA); OXPHOS WB antibody cocktail (#ab110411) from ABCAM (Cambridge, UK). Anti-mouse or anti-rabbit antibodies, conjugated with horseradish peroxidase, were used as secondary antibodies. Clarity western ECL Substrate (#1705061, BIO-RAD, Hercules, CA, USA) was used to detect bands, using ChemiDoc XRS (BIO-RAD, Hercules, CA, USA). Bands’ intensity was analyzed by ImageLab software (BIO-RAD, Hercules, CA, USA). Some blots were cut and probed with different antibodies for different proteins including β-Actin and α-Tubulin. In some figures, the β-Actin and α-Tubulin bands are the same as they belong to the same PDVF membrane. In some cases, to examine proteins of similar molecular weight, the PVDF membranes were subjected to a mild stripping protocol as recommended by ABCAM (Cambridge, UK).

### 2.6. ATP Assay

ATP was extracted following and readapting the protocol described by Shryock et al. [28]. Briefly, 3 × 10^5^ treated and untreated cells were resuspended in cold 2.5% trichloroacetic acid (TCA) and kept on ice for 60 min. Then, the samples were centrifuged at 2000 rpm for 5 min (4 °C), the supernatants collected in new cold tubes and 50 mM Tris-acetate buffer pH 7.75 added to a final TCA concentration of 0.1%. ATP was measured using the ENLITEN ATP assay system bioluminescence detection kit, according to the manufacturer’s instructions (FF2000, Promega, Madison, WI, USA) and bioluminescence read by Tecan Infinite-Pro 200 plate-reader. Samples from each independent experiment were analyzed in duplicate. The amount of ATP (pmol) was referred to a standard curve.

### 2.7. Flow Cytometry Analysis

For mitochondrial membrane potential quantification, cells were incubated with 200 nM TMRE (#T669, Invitrogen by Thermo Fisher Scientific, Whaltam, MA, USA) in serum-free medium for 20 min at 37 °C, trypsinized and resuspended in complete medium. Cells were then washed in PBS and fluorescence was measured as Median Fluorescence Intensity (MFI) at wavelength excitation/emission 549 nm/575 nm using BD FACSCanto™ II Flow Cytometry System (BD Bioscience, Franklin Lakes, NJ, USA).

### 2.8. Immunofluorescence and Microscopy

To obtain mitochondrial staining, cells were seeded in sterile Lab-Tek Chambered Coverglass #1 (#155383 Thermo Fisher Scientific, Whaltam, MA, USA) and treated with 30 nM MitoTracker Green (Invitrogen by Thermo Fisher Scientific, Whaltam, MA, USA) in serum-free medium for 15 min at 37 °C or with 200 nM TMRE (#T669, Invitrogen by Thermo Fisher Scientific, Whaltam, MA, USA) in serum-free medium for 20 min at 37 °C. Nuclei were labelled by incubating cells with 1 µg/mL Hoechst (Merck-Sigma #H6024, St. Louis, MI, USA) in serum-free media for 5 min. Fluorescence was analyzed using TCS SP8 confocal microscope (Leica, Wetzlar, Germany). Three-dimensional reconstruction was assessed using Leica LasX 3D software (Magnification 63X, Scale bar 5 μm).

### 2.9. Statistical Analysis 

All data are shown as mean ± SD of at least three independent experiments. Statistical significance was calculated using ANOVA with Dunnett’s post-test and significance shown as * *p* < 0.05, ** *p* < 0.01, *** *p* < 0.001. **** *p* < 0.0001 or *t-*test and shown as # *p* < 0.05. Statistical analysis was made using GraphPad Prism software, version 8.1.1 (GraphPad, San Diego, CA, USA). 

## 3. Results

### 3.1. Transcriptomic Analysis in CRC HCT116 Cells Treated with HTyr or OPWPs Extracts

In order to evaluate the modifications of the complete set of RNA transcripts, we treated CRC HCT116 cells for 72 h with HTyr or OPWPs (0.0154 mg mL^−1^) from Apulia (AE), Tuscany (TE) and Sicily (SE), whose composition has previously been reported [25]. Total RNA was extracted, purified and subjected to high throughput RNA-sequencing, as specified in the M & M section. From total 60,664 quantified transcripts, the OPWPs from AE, TE, SE and HTyr treatments were able to modify the expression of 2062, 1636, 3781 and 2146 genes, respectively (adjusted *p* < 0.05 and absolute fold change >1.5) (Figure 1A). Out of these, 1056 genes were found to be commonly and coherently altered following treatments; specifically, the expression of 571 genes was upregulated, while that of 485 genes was downregulated (Figure 1B). The remaining genes were influenced by single treatments (for details see Appendix A). 

To assess the functional impact of the observed variations in gene expression, we performed a pathway analysis using Gene Set Enrichment Analysis (GSEA) that determines whether a defined set of genes exhibits statistically significant differences between two biological conditions, in our case treated vs. untreated cells. HALLMARK (h.all.v2022.1) was used as the reference geneset. As shown in Figure 1C, we used two indexes, the enrichment score (ES) and the normalized enrichment score (NES), to evaluate and compare the analysis’ results between the genesets. The former reflects the degree to which a geneset is overrepresented at the top or bottom of a classified gene list; the latter takes into account the differences in the size of the geneset and the correlations between the geneset and the expression dataset. The analysis highlighted several significantly affected pathways (24, FDR *p* < 0.05); the enriched categories are illustrated in the distribution-plot, in which the size of the dots is proportional to the number of overlapping genes (Figure 1C and Appendix A). 

In detail, the first two most impacted and upregulated pathways were “E2F Target” and “G_2_/M Checkpoint”, suggesting that the main effect is a cell cycle block, in particular a G_2_/M phase transition arrest. E2F is an essential protein that regulates the cell cycle with a block at the G_2_/M phase through also the downregulation of Stathmin and AIM-1 [29,30]. Since a cell cycle arrest is strictly tied to restraining cell proliferation, E2F acts as a tumor suppressor. The third pathway identified was “MYC Target v1” that appears to be counter-intuitive since *c-MYC* is an oncogene. Indeed, a more detailed investigation of the genes revealed that nearly half of them are shared with other pathways and a significant proportion (35%) are downregulated, including *c-MYC*. This preliminary analysis supports the hypothesis that HTyr and the OPWPs extracts induce a cell cycle arrest at the G_2_/M phase transition with consequent growth inhibition, in line with similar results reported in the literature [24,31]. Consistent with these data, we recently showed that administration of the non-toxic dose of 0.0154 mg mL^−1^ HTyr, AE, TE and SE to HCT116 cells for 72 h, induced a cell cycle arrest at the G_2_/M phase and impaired cell proliferation. The treatment also elicited an anti-inflammatory response [25]. This result agrees with the present GSEA analysis that, among the most downregulated pathways, includes TNF-α signaling via NF-κB, TGF-β and IL2-STAT signaling, strongly supporting the anti-inflammatory role of HTyr and OPWPs extracts. Of note, among the downregulated pathways are comprised *KRAS*, Hypoxia, and Epithelial Mesenchymal Transition (Figure 1C and Appendix A), all related to the tumorigenic process, providing further support to the overall effect of impairing cell proliferation and tumor growth [32,33,34]. Interestingly, the fourth top upregulated pathway identified by the RNA-Seq analysis was OXPHOS, suggesting that HTyr or OPWPs extracts exposure affects mitochondrial respiration (Figure 1D). Indeed, by surveying the genes included in this pathway, we recognized 15 out of 57 genes, all upregulated, that are directly involved in the electron transport chain and, thus, with mitochondrial functionality. Therefore, we decided to investigate this pathway in depth by validating the effects of these compounds in a cell system in culture. 

### 3.2. HTyr and OPWPs Extracts Affect Mitochondrial Functions

To verify whether the genes belonging to the OXPHOS were actually influenced by HTyr or OPWPs extracts, we treated HCT116 and LoVo cells for 72 h with HTyr or OPWPs extracts. Cells were harvested and total protein extracts analyzed by Western blot for the expression of the five multi-protein complexes of the respiratory chain. HCT116 cells displayed an overall increase of all OXPHOS complexes for both HTyr and the OPWPs extracts (Figure 2A, left panel). LoVo cells showed similar results, more pronounced with AE and SE than with HTyr and TE (Figure 2A, right panel), demonstrating that the results are not restricted to a single cell line. Experiments carried out for shorter times (24 and 48 h) produced similar but less intense results and for this reason are not reported in this study [25].

The electron transport chain is functionally associated with the production of ATP. We, thus, measured the amount of ATP produced by the two cell lines treated as above. We found a stimulation by all OPWPs extracts and HTyr with respect to untreated cells; in particular, in HCT116 cells, we observed a 4-fold increase for SE and 3-fold for HTyr and AE treatments, respectively. In LoVo cells, we detected similar effects with a 3-fold increase in cells treated with HTyr, AE and SE (Figure 2B). Finally, the cells treated as above were stained with TMRE, tetramethylrhodamine ethyl ester, a selective dye to evaluate the effects on the mitochondrial membrane potential by flow cytometry. The results clearly show that the treatments significantly increase the membrane potential in both cell lines, with no notable differences among OPWPs extracts and HTyr (Figure 2C). All together, these results strengthen our hypothesis that OPWPs extracts positively influence mitochondrial functionality by increasing the expression of OXPHOS-related proteins, stimulating ATP production and enhancing the membrane potential. 

### 3.3. HTyr and OPWPs Extracts Promote Mitochondrial Biogenesis and Influence Dynamics

Next, we investigated whether the enhanced mitochondrial functionality is associated with increased mitochondrial mass. To this end, we analyzed the expression of TOM20, an integral protein of the outer mitochondrial membrane, commonly used as a marker of the overall mitochondrial amount in cells treated or not with HTyr or the OPWPs extracts. In HCT116 cells, we detected a significant increase in TOM20 protein after the treatments, with a slightly higher percentage for TE and SE (Figure 3A, left panel); in LoVo cells, the increase was more pronounced (Figure 3A, right panel). To further investigate if the enhanced TOM20 levels really reflected an increase in mitochondrial mass, treated and untreated cells were stained with MitoTracker Green, a specific cell-permeant mitochondrion-selective dye. By confocal microscopy analysis, we appreciated a fluorescent signal that was more intense in treated cells from both cell lines, strongly supporting the hypothesis that exposure to HTyr or the OPWPs extracts increases the mitochondrial mass (Figure 3B). These data suggest that the enhancement in respiratory potential and ATP production depends on the higher mitochondrial mass. We asked next if the augmented content was related to the formation of new mitochondria (biogenesis) or variations in the morphology of mitochondria through fusion and/or fission events (dynamics). To this purpose, we assessed the levels of the major proteins involved in these processes. FOXJ3 controls the very first step of the pathway by modulating PGC-1α either directly and through MEF2A and MEF2C, two members of the MEF2 family [27,35,36] (https://www.encodeproject.org/annotations/ENCSR285DHW, accessed on 20 October 2022). PGC-1α plays, in turn, a pivotal role in many mitochondrial activities, stimulating the expression of NRF1 and NRF2, two transcription factors which cooperate to activate nuclear and mitochondrial genes required for biogenesis and respiratory functions [17,37]. PGC-1α, in addition, stimulates the binding and activity of factors required for mitochondrial transcription factor A (TFAM) also involved in the replication and maintenance of mitochondrial DNA [20,38]. All together, these factors form a mutually self-regulatory and cross-regulatory network able to direct OXPHOS in muscle and in CRC cells [27,39]. In HCT116 cells, all treatments increased FOXJ3 protein, with TE and SE showing the strongest effect. The extracts enhanced FOXJ3 also in LoVo cells (Figure 3C). PGC-1α was stimulated by all treatments in both cell lines with a 2-fold increase compared to untreated cells (Figure 3D). Additionally, TFAM, the most downstream factor, showed in both cell lines a 2-fold increase in treated versus untreated cells (Figure 3E). Collectively, these data suggest that the increase in respiratory potential and mitochondrial mass following HTyr or the OPWPs extracts treatments are due to mitochondrial biogenesis. 

This process is, however, tightly linked to mitochondrial dynamics, whereby mitochondria change and rearrange their shape by altering the structures of the inner and outer membranes. 

Dynamics implies a balance between fusion and fission events: the former entails the fusion of two or more mitochondria to create a network; conversely, the latter entails mitochondrial fragmentation to generate multiple mitochondria with a heterogeneous shape. We thus assessed the levels of Mitofusin 1 and 2 (MFN1 and MFN2, respectively), two conserved transmembrane GTPases that coordinate the merging of the outer and inner membranes, as representative markers of fusion events. We also analyzed the Mito Fission Factor (MFF) that, instead, promotes fission. By Western blot analysis, we verified that MFN1 and MFN2 levels increased after the treatment with all the extracts compared to untreated cells (Figure 4A,B). Contrariwise, MFF was remarkably reduced by all treatments in both cell lines with a more evident effect for SE (Figure 4C), suggesting that HTyr and OPWPs extracts shift the balance between these two events towards fusion that, in turn, is associated with the formation of a mitochondrial network. To further appreciate if mitochondrial morphology was affected, treated cells were stained with TMRE and analyzed by confocal microscopy. SE was used as representative of all extracts and compared to untreated cells. Consistent with previous data, in treated versus untreated cells for both cell lines, we observed more abundant mitochondria with an elongated, tubular morphology connected in an intricate network (Figure 4D). These results support that HTyr and OPWPs extracts promote mitochondrial biogenesis and fusion leading to an increased number of functional mitochondria.

### 3.4. PPARγ Mediates the Effects of HTyr on Mitochondrial Biogenesis and Dynamics 

Next, we sought to unveil the mechanisms underlying the above-described effects of HTyr and OPWP extracts on mitochondrial structure and function. Data from the literature indicate that *PPARGC1* is the master gene that orchestrates a myriad of mitochondrial functions such as OXPHOS, fatty acids synthesis and oxidation, tricarboxylic acid cycle, in addition to the biogenesis already described [20,21,27]. Among the transcription factors regulated by this coactivator, PPARγ is involved in lipidic and energetic control [23,40]. In this context, we previously showed that PPARγ mediates the anti-inflammatory and antiproliferative effects of HTyr and OPWPs extracts in the same CRC cells [25]. Thus, we treated HCT116 and LoVo cells for 72 h with rosiglitazone (rosi) or GW9662, respectively a specific PPARγ ligand or a PPARγ irreversible inhibitor, either alone or in combination with HTyr. Total protein extracts from treated and untreated cells were analyzed by Western blot for the major proteins involved in mitochondrial biogenesis. Unexpectedly, FOXJ3, PGC-1α and TFAM were all induced by rosi to the same extent as HTyr. Exposure to GW9662 produced no variation and, interestingly, completely blocked the stimulation in combination with HTyr (Figure 5). 

These data demonstrate that PPARγ plays a crucial role in regulating mitochondrial biogenesis, as supported by the results of the treatments of HTyr alone or combined with the inhibitor. Subsequently, we asked whether PPARγ could also be implicated in the regulation of mitochondrial dynamics. To uncover this aspect, we treated cells as above and evaluated mitochondrial fusion and fission markers. As shown in Figure 6A,B, in both cell lines, MFN1 and MFN2 were stimulated by about 2-fold after both HTyr and rosi treatment, while no effects were observed upon GW9662. Moreover, also in this case, the stimulation by HTyr was abrogated in the combined treatment with GW9662. Conversely, the exposure to HTyr or rosi caused a diminution of MFF that was abrogated by the combined treatment with GW9662 (Figure 6C). All together, these results indicate that HTyr and OPWPs extracts regulate mitochondrial biogenesis and the fusion events underlying mitochondrial dynamics through PPARγ. 

## 4. Discussion

In this study, we provide evidence that HTyr and OPWPs extracts upregulate the OXPHOS pathway in CRC cell lines. RNA-sequencing analysis of HCT116 cells treated with HTyr or the OPWPs extracts, followed by GSEA analysis of the major impacted pathways, identified E2F, G_2_/M phase transition cell cycle block and *c-MYC* targets, as the first three top influenced ones. E2F is a pivotal regulator of the cell cycle and its upregulation leads to a specific arrest. Coherently, the G_2_/M pathway is upregulated so to elicit a robust reduction of cell proliferation. The *c-MYC* target pathway is also upregulated and, despite being an oncogene, a detailed survey shows that many genes included in this list are involved in restraining cell growth. The negative impact of HTyr and OPWPs extracts on the cell cycle and on the ensuing cell proliferation has already been investigated and reported in the literature [24,31]. Besides, we have recently shown, in the same CRC cell model, that HTyr and OPWPs extracts cause a cell cycle arrest at the G_2_/M phase transition, impairment of the tumorigenic potential, and induction of apoptosis, in line with the results presented here [25]. Otherwise, less is known about the impact of HTyr and OPWPs extracts on the OXPHOS, which emerged as the fourth subsequent top pathway identified in the RNA sequencing analysis. Fifteen genes of the list were upregulated, all involved in the electron chain transport and ATP synthesis, suggesting that HTyr and OPWPs extracts could promote mitochondrial respiration. Since the mitochondrial membrane potential is also increased, we assumed that HTyr and OPWPs positively influence overall mitochondrial functionality.

Enhancement of mitochondrial functionality correlates with an increase in the mass mainly as the result of mitochondrial biogenesis. This process implies the production of new functional organelles in response to exogenous challenges that demand more energy [15,16,17] and is regulated by a series of factors interconnected in a network. The mitochondrial content in a cell is also controlled by mitochondrial dynamics that implies fusion and fission events. Upon fusion, the content of partially damaged mitochondria is mixed with components from healthy mitochondria, providing a form to mitigate the stress and a marker of functionality [18,19,41]. Fission, instead, generates new mitochondria and contributes to quality control by enabling the removal of damaged mitochondria [41,42]. Biogenesis and dynamics finely tune mitochondrial homeostasis, maintaining the proper mitochondrial structure and functionality, distribution and movement when cells experience stress, contributing to the cellular equilibrium [16]. We clearly show that HTyr and the OPWPs extracts promote mitochondrial biogenesis in both the analyzed CRC cell lines by stimulating the expression of the major factors. Moreover, they affect the dynamics, tilting the ratio between fusion and fission events towards fusion, as documented by the increase of MFN1 and MFN2 and by the reduction of MFF. Consistently, the mitochondria of treated cells acquire a more elongated shape resulting in an interconnected network, a marker of functionality. Data from the literature report that mitochondrial biogenesis is stimulated by HTyr and oleuropein, its natural precursor, in avian muscle cells leading to an increased mitochondrial mass through activation of PGC-1α [43]. In mouse 3T3L1preadipocytes, HTyr forces the terminal differentiation into adipocytes, reducing the size and the number of the fat droplets accumulated during the differentiation process [44]. In fish liver cells, HTyr reverts the mitochondrial dysfunction due to lipid accumulation by enhancing the respiratory potential and regaining functional mitochondria [45]. Additionally, in a number of in vivo and in vitro models of neurodegenerative disorders, HTyr restores the mitochondrial respiratory potential [46]. Thus, HTyr/oleuropein stimulate mitochondrial biogenesis and functionality. Contradictory results have, however, been reported in other cell systems so that a final and univocal conclusion cannot be drawn [47].

The data reported here shed some light on this disputed question. First of all, we demonstrate that OPWPs extracts positively affect mitochondrial respiration and ATP synthesis by fostering biogenesis and dynamics in CRC cells, as HTyr. Moreover, for the first time, to the best of our knowledge, we provide evidence that HTyr exerts its effects through PPARγ that acts as a specific receptor. The upregulation of the major factors involved in biogenesis upon treatment with the selective PPARγ ligand rosiglitazone to an extent similar to HTyr strongly supports this notion, further confirmed by its impairment subsequent to an irreversible PPARγ inhibitor. That PGC-1α is modulated by HTyr and rosiglitazone suggests a feed-forward regulatory loop between these two factors to control mitochondrial functions, according to the following model. Upon ligand binding (HTyr or rosiglitazone), PPARγ becomes transcriptionally active, recognizes the PPAR responsive elements (PPREs) in the promoter regions of target genes and stimulates their transcription. *FOXJ3* is one of the targets as it responds to PPARγ with an increase of the protein, as shown here and reported in the ENCODE project (https://www.encodeproject.org/annotations/ENCSR408JQP/, accessed on 20 October 2022). FOXJ3, in turn, modulates *PPARGC1A* either directly (https://www.encodeproject.org/annotations/ENCSR285DHW/, accessed on 20 October 2022) and through members of the MEF2 family of transcription factors (MEF2A and MEF2C) [35,36]. *PPARGC1A* is also likely to be regulated by PPARγ as PPREs have been reported in its promoter region (https://www.encodeproject.org/annotations/ENCSR408JQP/, accessed on 20 October 2022). The resulting elevated PGC-1α protein, in turn, acts as a coactivator to the transcription factors NRF1 and NRF2 on the promoter of TFAM, stimulating its expression [27,37]. Further activation can be obtained by the direct binding of FOXJ3 to *TFAM* promoter as well as to that of *TOMM20* (https://www.encodeproject.org/annotations/ENCSR285DHW/, accessed on 20 October 2022, see Appendix A). This regulatory circuit can explain the results reported here: stimulating or restraining PPARγ positively or negatively affects the factors implicated in mitochondrial biogenesis. 

A similar regulatory circuit appears to control mitochondrial dynamics, as both HTyr and rosiglitazone stimulate the expression of MFN1, MFN2 and reduce MFF. MFN1 is, in fact, the target of transcription factors of the FOXO family and, presumably, also of FOXJ3 [48]. MFN2 has been reported to be a PGC-1α and PPAR target, as distinct PPREs have been found in its promoter region [49]. No binding sites for PPARγ have been found in the regulatory regions of other genes. That the fusion and fission proteins are coordinately up and downregulated, respectively, suggests that PPARγ orchestrates these events. Notably, the effects of HTyr and OPWPs extracts presented here are in line with those we previously reported, showing that the PPARγ/PGC-1α axis mediates the impairment of cell proliferation and induction of apoptosis in the same model system [25]. 

## 5. Conclusions

We provide strong evidence that, in CRC cell lines, HTyr and OPWPs extracts positively influence the OXPHOS pathway and promote mitochondrial biogenesis and dynamics. This results in an enhanced mass of functional mitochondria organized in an intricate network to sustain cell metabolism. Mechanistically, PPARγ mediates these effects through a regulatory loop with PPGC-1α. These results support the assessment of the effects in in vivo models and may pave the way to use these compounds as adjuvants in cancer therapy and nutraceuticals for human wellness. 

## Figures and Tables

**Figure 1 cells-11-03992-f001:**
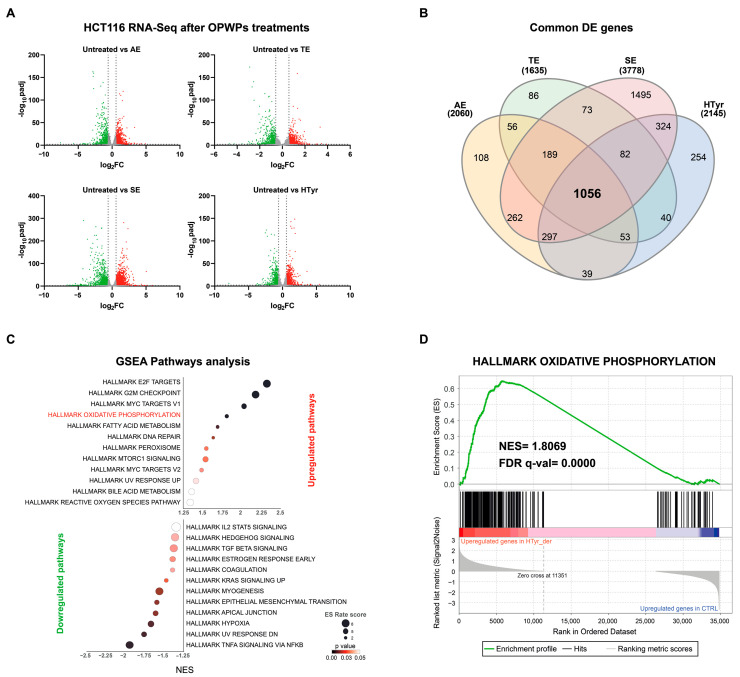
RNA-Seq analysis on HCT116 cells untreated or treated with HTyr or the OPWPs extracts. (**A**) RNA-Seq analysis was carried out on HCT116 cells treated or not with HTyr or selected OPWPs extracts, namely AE, TE, SE, respectively. The Volcano plots illustrate the Differentially Expressed Genes (DEGs) selected by applying a *p* adj < 0.05 and Log_2_FC > 0.585 or Log_2_FC < −0.585. (**B**) The Venn Diagram shows the common DEGs after treatments, out of which 571 were upregulated and 485 downregulated. GSEA analysis was performed on DEGs and HALLMARK (h.all.v2022.1) was used as reference geneset. (**C**) The Enrichment Score (ES) and Normalized Enrichment Score (NES) were used to generate a ranking of the top inferred pathways. OXPHOS emerged as the fourth most influenced pathway by HTyr or OPWPs treatments. (**D**) The distribution plot depicts the enriched genes of the OXPHOS pathway, most of which were upregulated. The size of the dots is proportional to the number of overlapping genes.

**Figure 2 cells-11-03992-f002:**
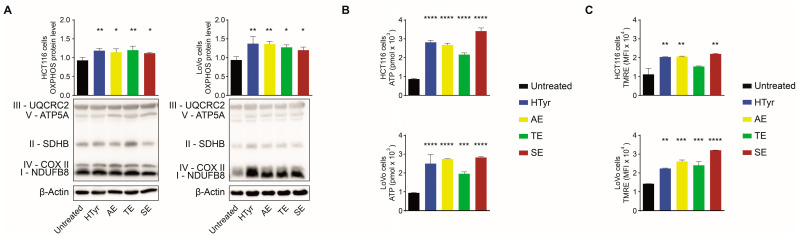
HTyr and OPWPs extracts affect mitochondrial functions. (**A**) Representative immunoblots of lysates of HCT116 and LoVo cells untreated or treated with HTyr or OPWPs extracts for 72 h for assessing the proteins ATP5A, UQCRC2, SDHB, COX II, and NDUFB8 belonging to the respiratory chain complexes V, III, II, IV, and I, respectively. The histograms report the overall OXPHOS protein quantification normalized to β-Actin or α-Tubulin, used as loading controls, and show the mean ± SD of three independent experiments. (**B**) ATP levels measured in HCT116 and LoVo cells treated as in (**A**) and expressed as picomoles (pmol). (**C**) Flow cytometry analysis of the cells treated as in (**A**) and stained with TMRE to evaluate the mitochondrial membrane potential calculated as Median Fluorescence Intensity (MFI) of two independent experiments. Statistical significance was considered when * *p* < 0.05, ** *p* < 0.01, *** *p* < 0.001, or **** *p* < 0.0001 (ANOVA with Dunnett’s post-test).

**Figure 3 cells-11-03992-f003:**
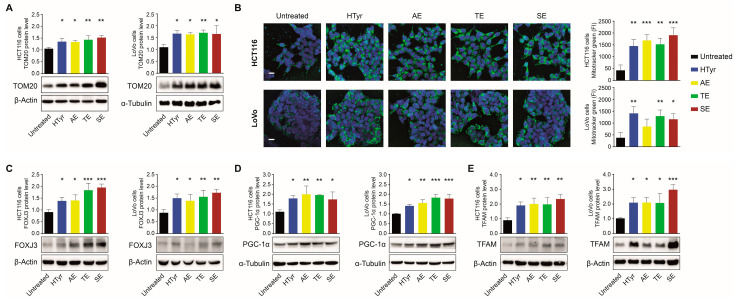
HTyr and OPWPs extracts influence mitochondrial abundance in a CRC cell model system. (**A**) Western blot analysis of TOM20 in HCT116 and LoVo cells treated with HTyr and OPWPs for 72 h. Protein level was calculated after normalization to β-Actin or α-Tubulin, used as loading controls. Three independent experiments were performed. (**B**) Confocal fluorescence microscopy images of the same cells treated as in (**A**) stained with MitoTracker Green to evaluate the mitochondrial mass. Hoechst was used to stain nuclei (blue). Magnification 63X, Scale bar 5 μm. The histograms on the right illustrate the values of the mitochondrial mass calculated as Fluorescence Intensity (FI) with the ImageJ software, as described in Materials and Methods; immunoblot analysis of FOXJ3 (**C**), PGC-1α (**D**), and TFAM (**E**) as mitochondrial biogenesis markers in HCT116 and LoVo cells treated as in (**A**). The histograms illustrate the results expressed as protein levels after normalization to α-Tubulin or β-Actin, used as loading controls, and show the mean ± SD of three independent experiments. Statistical significance was considered when * *p* < 0.05, ** *p* < 0.01 or *** *p* < 0.001 (ANOVA with Dunnett’s post-test).

**Figure 4 cells-11-03992-f004:**
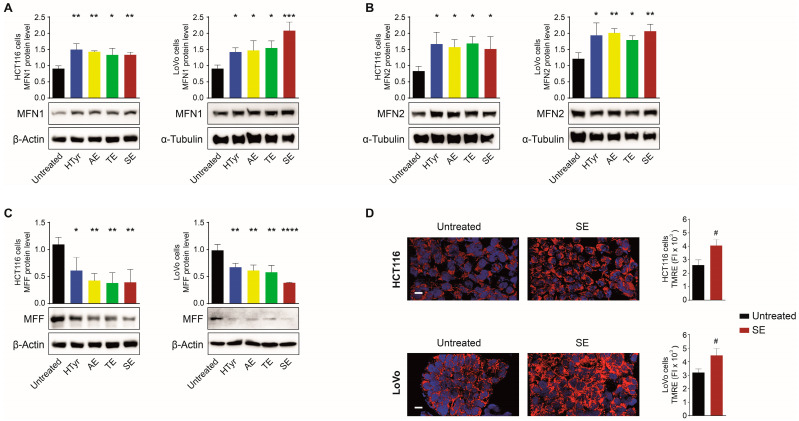
HTyr and OPWPs extracts influence mitochondrial dynamics. Protein lysates from HCT116 and LoVo cells treated with HTyr or OPWPs extracts were analyzed by Western blot for MFN1 (**A**), MFN2 (**B**) and MFF (**C**). Protein levels were calculated after normalization to β-Actin or α-Tubulin, used as loading controls. Three independent experiments were performed; the bands corresponding to β-Actin in (**A**,**C**) for HCT116 cells and shown here are the same, since the MFN1 and MFF proteins were detected on the same filter. (**D**) Representative confocal fluorescence microscopy images of HCT116 and LoVo cells treated with SE for 72 h and stained with TMRE to evaluate mitochondrial morphology. The histograms on the right illustrate the values of the mitochondrial mass calculated as Fluorescence Intensity (FI) with the ImageJ software, as described in Materials and Methods. Statistical significance was considered when * *p* < 0.05, ** *p* < 0.01, *** *p* < 0.001, or **** *p* < 0.0001 (ANOVA with Dunnett’s post-test) for (**A**–**C**) and # *p* < 0.05 (*t-*test) for (**D**).

**Figure 5 cells-11-03992-f005:**
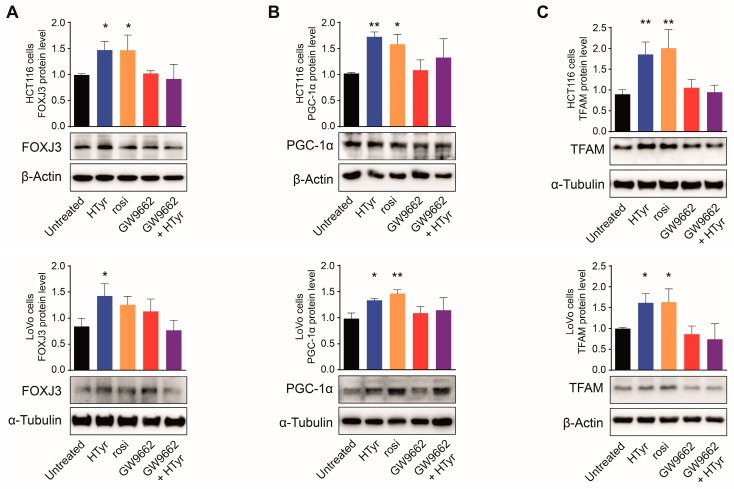
PPARγ mediates the effects of HTyr on mitochondrial biogenesis. Representative immunoblots of FOXJ3 (**A**), PGC-1α (**B**), and TFAM (**C**), as markers of mitochondrial biogenesis in HCT116 and LoVo cells untreated or treated with HTyr, rosi, GW9662 or the GW9662 + HTyr combined treatment. The results are expressed as protein levels after normalization to β-Actin or α-Tubulin, used as loading controls, from three independent experiments. Statistical significance was considered when * *p* < 0.05 or ** *p* < 0.01 (ANOVA with Dunnett’s post-test).

**Figure 6 cells-11-03992-f006:**
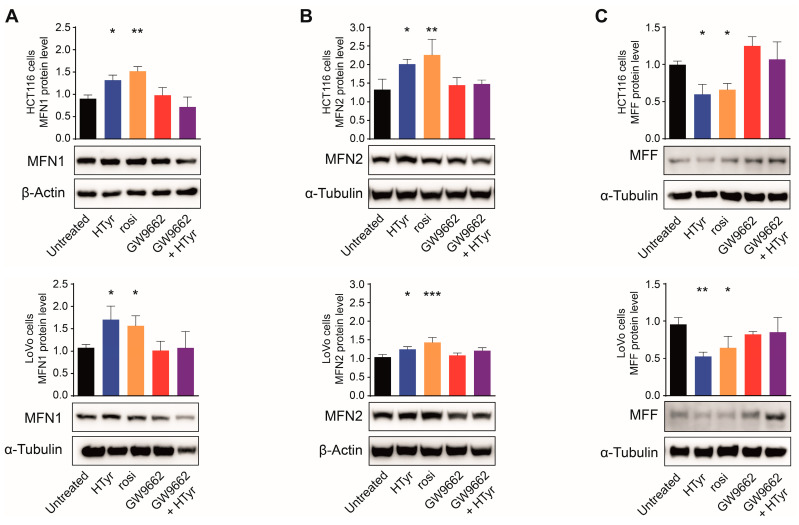
PPARγ mediates the effects of HTyr on mitochondrial dynamics. Western blot analysis of MFN1 (**A**) and MFN2 (**B**) as representative of fusion proteins, and MFF (**C**) as representative of fission proteins, in HCT116 and LoVo cells untreated or treated with HTyr, rosi, GW9662 or the GW9662 + HTyr combined treatment. The histograms represent protein levels after normalization to β-Actin or α-Tubulin, used as loading controls, and show the mean ± SD of at least three independent experiments. Statistical significance was considered when * *p* < 0.05, ** *p* < 0.01 or *** *p* < 0.001 (ANOVA with Dunnett’s post-test).

## Data Availability

The normalized data supporting the conclusions of this article are available in the Appendix A.

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
