# Peer review of "Transcriptomic Analysis of Colorectal Cancer Cells Treated with Oil Production Waste Products (OPWPs) Reveals Enrichment of Pathways of Mitochondrial Functionality"

_cells, 2022, doi:10.3390/cells11243992_

Round 1

Reviewer 1 Report

This is a very well designed and executed study with important information on the potential usefulness of supplementing the treatment for CRC with OPWPs. The paper is of good quality and have only minor comments.

Introduction: End of last paragraph the word “dynamism” appears. This is an incorrect terms as the authors should refer to “dynamics” and not “dynamism”. The authors need to have this term corrected throughout the text.

Methods: 

a) What was a Log2FC of plus or minus 0.585 selected in the GO analysis of DEG genes instead of e.g 0.6 or 0.7? Is it matter of convenience set arbitrarily or there is a reason mentioned ion the according literature?

b) Why was the 72 hours time point selected instead of 48 or 96? Please provide a rationale.  

Discussion:

Please revise English, in some cases there are minor mistakes (e.g. these evidences).

References

Ref 48: the name of the journal should be abbreviated

Reviewer 2 Report

In this study, the authors aimed to examine the transcriptional changes occurring in colorec-tal cancer (CRC) cells following the treatment with OPWPs extracts or the major constitu-ent HTyr. They focused on Oxidative Phosphorylation (OXPHOS) since it was among the top four most differentially regulated pathways. It was shown that OPWPs extracts and HTyr greatly affected mitochondrial respiration and mass by promoting biogenesis and dyna-mism. Mechanistically, these effects were mediated by the PPAR and PGC-1 axis through a feed-forward regulatory loop. Thus, OPWPs extracts and HTyr promoted mitochondrial functionality that added to their antiproliferative and proapoptotic activities. The theme is clear, the experimental design is reasonable. However, there are still several points which need to be revised before accepting for publication.

1.      The author only did in vitro experiments based on human CRC-derived cell lines, HCT116 and LoVo. Is the evidence sufficient? Do the authors consider in vivo animal validation?

2.      How about the dosage evidences, as the authors mentioned that “In all experiments, HCT116 cells were treated with 0.0154 mg mL−1 HTyr or 0.0154 mg mL−1 OPWPs extracts for 72 h; LoVo cells were treated with 0.0231 mg mL−1 HTyr or 0.0231 mg mL−1 OPWPs extracts for 72 h.”?

3.      Some blots of Actin expression in WB are irrugular.

4.      Is it reasonable to mark P value “≤ , such as * p 0.05, ** p 0.01, *** p 0.001. **** p 0.0001 ? It is always using <.

5.      For all WB bands, how many samples in each group? How aout the repeated times? It needs to be noted in figure legends.

6.      It is suggested horizontal arrangement in Figure 5 and Figure 6.

7.      Literatures in recent three years may be cited, especially in introducing the latest developments in this field.

8.      The author did not well explain why they paid attention to OPWPs on the function of mitochondria.

9.      In addition to these indicators discussed by the authors, whether there are any other morphological evidences or results for evaluating the mitochondrial function?

Round 2

Reviewer 2 Report

The points have been addressed.